# MEASURING IN-CONTEXT COMPUTATION COMPLEXITY VIA HIDDEN STATE PREDICTION

**Vincent Herrmann** *
IDSIA/USI/SUPSI
Lugano, Switzerland

**Róbert Csordás**
Stanford University
Stanford, USA

**Jürgen Schmidhuber**
KAUST, IDSIA/USI/SUPSI
Thuwa, Saudi-Arabia / Lugano, Switzerland

## ABSTRACT

Detecting when a neural sequence model does "interesting" computation is an open problem. The next token prediction loss is a poor indicator: Low loss can stem from trivially predictable sequences that are uninteresting, while high loss may reflect unpredictable but also irrelevant information that can be ignored by the model. We propose a better metric: measuring the model's ability to predict its own future hidden states. We show empirically that this metric—in contrast to the next token prediction loss—correlates with the intuitive interestingness of the task. To measure predictability, we introduce the architecture-agnostic "prediction of hidden states" (PHi) layer that serves as an information bottleneck on the main pathway of the network (e.g., the residual stream in Transformers). We propose a novel learned predictive prior that enables us to measure the novel information gained in each computation step, which serves as our metric. We show empirically that our metric predicts the description length of formal languages learned in-context, the complexity of mathematical reasoning problems, and the correctness of self-generated reasoning chains.

## 1 INTRODUCTION

Neural sequence models—especially large language models (LLMs)—have demonstrated remarkable capabilities, such as solving complex reasoning tasks Lewkowycz et al. (2022); Wei et al. (2022); Zelikman et al. (2022) and performing in-context learning Brown et al. (2020); Zhao et al. (2021); Liu et al. (2022); Kirsch et al. (2022). Recent progress in mechanistic interpretability has offered glimpses into how these models process information internally: for instance, by identifying specific induction heads Olsson et al. (2022) or $n$-gram mechanisms Akyürek et al. (2024) that capture non-trivial aspects of the underlying computation. Nevertheless, quantifying in a general way *when* and *how much* truly interesting or complex computation is happening within a model remains challenging.

A natural first instinct is to use next-token prediction loss to gauge complexity: Computations that yield high loss might appear "hard," whereas those with low loss might be "easy". However, this notion of hardness does not necessarily reflect the complexity or "interestingness" of underlying computation, see e.g.,

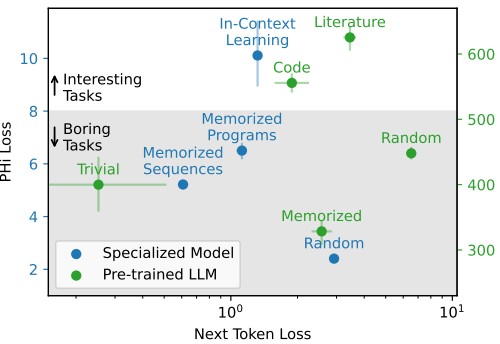

Figure 1: Interesting tasks, like in-context learning, or modeling of code and literature, exhibit high hidden state prediction (PHi) loss, while boring or trivial tasks, such as retrieving memorized sequences or modeling random structureless data, show low PHi loss. Next token loss provides no meaningful insight into task complexity. Results for a specialized transformer model (blue) and a pre-trained LLM (green), with PHi loss scales differing due to hidden state size. See Sections 3.1.1 and 3.2.1 for details.

---

*Correspondence to `vincent.herrmann@idsia.ch`

Schmidhuber (1991a;b). For example, a random sequence has a high next token prediction loss, but at the same time, the model does not have to compute anything because there is nothing that it can do to help the predictions. Hence, next-token prediction alone fails to capture the complexity of in-context computation performed by the model.

We therefore propose a new metric: *hidden-state predictability*. Concretely, we measure how well the model can predict its own future hidden states, rather than future tokens. To achieve this, we introduce a PHi ("**P**rediction of **Hi**dden States") layer, which imposes an information bottleneck on the hidden representations, and predicts them autoregressively. The layer is placed in the information pathway of the autoregressive sequence model. Under this scheme, the activations flowing through the PHi layer have to be compressed—information is included in the hidden state only if it is expected to be relevant for generating future tokens. The model is encouraged to encode the "in-context program" needed for sequence prediction in the hidden states. The hidden state prediction loss (PHi loss) allows us to quantify the complexity, or description length, of this program. Our approach offers several advantages: (i) It is architecture-agnostic and can be inserted in a wide range of autoregressive models, including transformers (Vaswani et al., 2017) and RNNs (Elman, 1990; Jordan, 1986; Hochreiter & Schmidhuber, 1997). (ii) Model and PHi layer can be jointly trained from scratch, or the PHi layer can be added post-hoc to pretrained LLMs. (iii) It provides a way to measure when the model is doing complex computation by examining the unpredictability—and hence information content—of the hidden states.

We provide empirical evidence, using both smaller-scale transformers/RNNs and large pre-trained language models, that our hidden-state unpredictability metric correlates with task complexity in various domains. Concretely, we show the following:

- Tasks that require no interesting data-processing, such as reciting memorized sequences, executing memorized programs in-context, or modeling random noise, are associated with low PHi loss. Interesting tasks, such as in-context learning, or modeling complex natural language and code, yield high PHi loss.

- For in-context learning formal stochastic languages, PHi loss is predictive (beyond next token loss) of the complexity of the language.

- For step-by-step solutions of math problems, PHi loss is predictive (again, beyond next token loss) of the hardness of the problem.

- When an LLM itself generates step-by-step solutions to math problems, solutions with high PHi loss have a significantly increased chance of being correct.

Taken together, these insights shed light on the internal computations of powerful sequence models and offer a principled tool for detecting "interesting" in-context behaviors. The remainder of this paper is organized as follows: in Section 2, we introduce the PHi layer and describe how it is used to measure predictability, and how it is integrated into an autoregressive model. In Section 3, we demonstrate the effectiveness of our approach on a variety of tasks, showcasing how hidden-state predictability correlates with task complexity. Section 4 reviews related work before we discuss limitations and avenues for further research in Section 5. We conclude with Section 6.

## 2  METHOD

In the following, Section 2.1 introduces our PHi layer, which serves as a basis of our novel metric. Then, we build on this layer to introduce our proposed quantification of "interestingness" in Section 2.2. In Section 2.3, we discuss the training of our layer, and we finish with Section 2.4 by discussing the connection between our proposed PHi loss and the next token prediction loss.

### 2.1  PREDICTION OF HIDDEN STATES

We consider a standard autoregressive sequence model $\pi_\theta$, with parameters $\theta$, so that

$$p_\theta(x_{1:T}) \;=\; \prod_{t=1}^{T} \pi_\theta(x_t \mid x_{<t}).$$ (1)

In practice, $\pi_\theta(\cdot \mid x_{<t})$ can be realized by any neural sequence model such as a decoder-only transformer or an RNN. As long as $\pi_\theta$ has at least two sequence layers, we can split this model into a *bottom* set of layers and a *top* set of layers.

**Bottom Layers:** The function $B_\beta$, with parameters $\beta$, processes the sequence $(x_1, \ldots, x_t)$ into a hidden representation $h_t = B_\beta(x_1, \ldots, x_t)$. One can think of $B_\beta$ as the first few layers of the network.

**Top Layers:** The top (remaining) layers, denoted $T_\tau$, with parameters $\tau$, use hidden representations $\{h_1, \ldots, h_{t-1}\}$ to parameterize the next-token distribution: $\pi_\theta(x_t \mid x_{<t}) = T_\tau(x_t | h_1, \ldots, h_{t-1})$.

**PHi Layer:** We introduce a new layer that has distinct functions: It (a) introduces an information bottleneck on the current hidden state $h_t$ and (b) predicts *future hidden states*. We call this a PHi layer (see Figure 2). It is sandwiched between $B_\beta$ and $T_\tau$, and transforms $\{h_1, \ldots, h_t\}$ into a new sequence of representations $\{h'_1, \ldots, h'_t\}$. The layer is inspired by variational autoencoders (Kingma & Welling, 2014), but instead of a fixed prior, we use an autoregressive, learned prior, with access to the past states. First, we introduce latent variables $\{z_1, \ldots, z_t\}$ at each time step. The learned encoder $q_\psi(z_t \mid h_t)$ maps each hidden state $h_t$ to a posterior over $z_t$. The autoregressive, learned prior $p_\chi(z_t \mid z_1, \ldots, z_{t-1})$ generates a distribution over $z_t$ based on the previous history of latent variables. Thus, the prior serves as a predictor of future hidden states. In practice, the

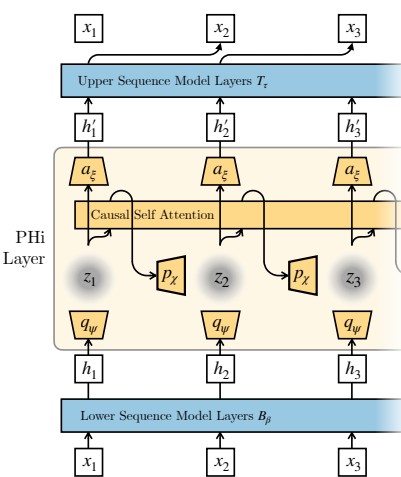

Figure 2: The structure of our PHi layer. It can be inserted in the middle of any next token prediction architecture, and it reconstructs its hidden states through an information bottleneck. It consists of posterior encoder $q_\psi(z_t \mid h_t)$ that predicts the latent code $z_t$, decoder $a_\xi$ used to reconstruct the hidden state $h'_t = a_\xi(z_t)$, and a learned autoregressive prior $p_\chi(z_t \mid z_1, \ldots, z_{t-1})$ predicting $z_t$ from the past latent codes. We propose to use the KL divergence between the posterior and the prior to quantify the complexity of the "in-context computation" performed by the model.

prior can be implemented by any causal sequence processing module, such as self-attention combined with a step-wise multi-layer percepetron (MLP). A decoder $a_\xi$, parametrized by a small neural network, is used to reconstruct the hidden states $h'_t = a_\xi(z_t)$ from the latents $z_t \sim q_\psi(\cdot \mid h_t)$.

The PHi layer can easily be inserted into most autoregressive sequence modeling architectures. The sequence model $\pi_\theta$ can be trained jointly with the PHi layer, or alternatively can be pre-trained, optimizing only the new PHi layer while holding the model's weights fixed.

## 2.2 Quantifying "Interestingness"

We propose to quantify interestingness by measuring the amount of information gained from the posterior $q_\psi(z_t \mid h_t)$ over the prior $p_\chi(z_t \mid z_1, \ldots, z_{t-1})$. This is precisely the KL divergence $D_{\mathrm{KL}}(q_\psi \| p_\chi)$. We define the PHi loss as the the information gained in a given timestep $t$:

$$L_{\mathrm{PHi}}(t) = D_{\mathrm{KL}}\Big(q_\psi(\cdot \mid h_t) \,\big\|\, p_\chi(\cdot \mid z_1, \ldots, z_{t-1})\Big) \tag{2}$$

Intuitively, this measures the nats of novel information learned in timestep $t$ that is not predictable from the past. We propose to use this quantity as the measure of the interestingness of the computation performed in a specific timestep.

When quantifying the interestingness of an entire sequence, we calculate the mean of the above metric over all tokens:

$$L_{\mathrm{PHi}} = \frac{1}{T} \sum_t^T L_{\mathrm{PHi}}(t) \tag{3}$$

We report this metric in our experiments in Section 3.

## 2.3 TRAINING OBJECTIVE

We train the entire system (bottom layers $B_\beta$, top layers $T_\tau$, and PHi layer) by combining two losses. First, we need to ensure that ensures that the *transformed* hidden states $\{h'_1, \ldots, h'_{t-1}\}$ still suffice to predict the token $x_t$. We do this by using the standard negative log-likelihood next token prediction loss used in language modeling:

$$L_{\mathrm{NLL}}(t) \; = \; -\log T_\tau\big(x_t \mid h'_1, \ldots, h'_{t-1}\big) \tag{4}$$

In parallel, we also minimize the PHi loss (Eq. 2). This encourages each $z_t$ to be predictable from its predecessors $z_{<t}$ by minimizing the KL divergence between the posterior and the autoregressive prior. Thereby, the "unnecessary" information that is not expected to be relevant for future predictions is discouraged in $z_t$.

The total loss for time step $t$ is:

$$L(t) \; = \; L_{\mathrm{NLL}}(t) \; + \; L_{\mathrm{PHi}}(t) \tag{5}$$

In effect, the model is incentivized to maintain enough information in the latent $z_t$ (and hence $h'_t$) to predict the next token, but not so much as to deviate from the learned autoregressive prior over $z$. This enforces an information bottleneck on the hidden states while preserving good predictive performance on the original sequence task.

We treat the latent variables $\{z_t\}$ in a fashion similar to a variational autoencoder (VAE) by sampling them from the approximate posterior $q_\psi(z_t \mid h_t)$. To enable gradient backpropagation through these samples, we use the standard reparameterization trick Kingma & Welling (2014); Rezende et al. (2014). Specifically, the posterior is modeled as a Gaussian with mean $\mu_\psi(h_t)$ and diagonal covariance $\sigma_\psi(h_t)^2$, we draw an auxiliary noise variable $\epsilon \sim \mathcal{N}(0, I)$ and set $z_t = \mu_\psi(h_t) + \sigma_\psi(h_t)\,\epsilon$. This approach provides a low-variance gradient estimator for updates to the parameters $\psi$. Consequently, the entire network (including bottom layers $B_\beta$, the prior $p_\chi$, and the output transform $a_\xi$) can be trained end to end via standard stochastic gradient descent.

## 2.4 THE CONNECTION BETWEEN $L_{\mathrm{NLL}}$ AND $L_{\mathrm{PHi}}$

Because the token sequence $x_{1:T}$ is modeled autoregressively by equation 1, we can use a standard entropy-coding technique (e.g. arithmetic coding, compare Deletang et al. (2024)) to encode $x_{1:T}$ in

$$\sum_{t=1}^{T}\big(-\log \pi_\theta(x_t \mid x_{<t})\big) \; = \; \sum_{t=1}^{T} L_{\mathrm{NLL}}(t) \quad \text{nats.}$$

The PHi layer allows us to proceed similarly with the hidden state sequence, or more precisely with the sequence of posteriors $\{q_\psi(\cdot|h_t), t \in 1, \ldots T\}$ over the latent variables $z_{1:T}$. The number of nats required to encode the entire latent sequence is

$$\sum_{t=1}^{T} D_{\mathrm{KL}}\Big(q_\psi(\,\cdot\mid h_t) \,\big\|\, p_\chi(\,\cdot\mid z_{<t})\Big) \; = \; \sum_{t=1}^{T} L_{\mathrm{PHi}}(t).$$

This description length of the hidden states allows us to quantify the amount of information that the model is extracting from the input sequence and is used to predict the remainder of the sequence. Note that the token $x_{\hat{t}}$ at any particular position $\hat{t}$ alone can be encoded in $L_{\mathrm{NLL}}(\hat{t}) + \sum_{t=1}^{\hat{t}-1} L_{\mathrm{PHi}}(t)$ nats by first encoding the latent sequence $z_{1:\hat{t}-1}$ and then encoding $x_{\hat{t}}$ given $T_\tau(\cdot|h'_1, \ldots h'_{\hat{t}-1})$ without explicitly encoding the past sequence of tokens.

## 3 EXPERIMENTS

We split the experiment section into two parts: Section 3.1 covers smaller scale sequence models that are trained jointly with the PHi layer from scratch. Section 3.2 investigates inserting a PHi layer into a pre-trained LLM.

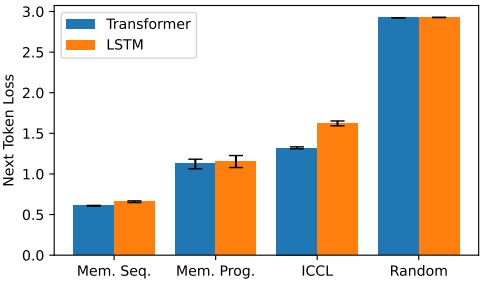 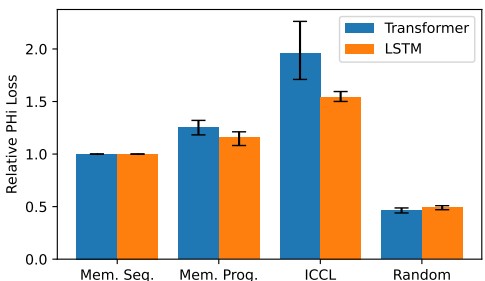

Figure 3: **Next-token prediction loss** on each of the four tasks, for both the Transformer and the LSTM. Memorized tasks yield the lowest loss, random is the highest, and the in-context language learning task is intermediate. Bootstrapped mean with 95% confidence intervals across 10 runs.

Figure 4: **PHi loss**, relative to the performance on the memorized sequences, for the same four tasks. In-context language learning shows a significantly higher PHi loss than the other tasks, indicating that the model is performing non-trivial computation in its hidden states to infer the unknown automaton. Bootstrapped mean with 95% confidence intervals across 10 runs.

## 3.1 EXPERIMENTS WITH FULLY TRAINED SEQUENCE MODELS

We begin our evaluation by training autoregressive sequence models from scratch with the proposed hidden-state prediction framework. Specifically, we consider two architectures: a decoder-only transformer and an LSTM. The transformer has 12 layers and a hidden state size of $768$. The LSTM has 2 layers and the same hidden state size. In each network, we insert a single PHi layer in the middle of the model. Exact hyperparameters and further details can be found in Appendix B. Our primary goal is to explore how the PHi approach responds to different tasks, especially those that require non-trivial in-context computation.

**Background: In-Context Language Learning with Probabilistic Finite Automata** We adopt the in-context language learning setup proposed by Akyürek et al. (2024), where each *problem instance* is defined by a probabilistic finite automaton (PFA). A PFA has a set of states and directed edges, each labeled with a token from some vocabulary. To generate an example from a PFA, we start at a random initial state and randomly sample an outgoing edge (with uniform probability among that state's edges). We emit the associated token, move to the edge's target state, and repeat for a random length of 10–50 tokens. In in-context language learning, the model is presented with multiple examples generated from the same unknown PFA. To achieve good next-token predictions on subsequent examples, the model must infer the structure of the PFA in-context.

We quantify the theoretical complexity of the "program" that must be learned in-context by calculating the number of bits necessary to encode the PFA. Let a PFA $A$ have $n$ states and $m$ edges, each labeled with a token from a chosen subset of size $v$ out of a total vocabulary of size $V$. We encode $A$ by (1) specifying $\log_2 \binom{V}{v}$ bits to choose which $v$ tokens are used; (2) listing $m$ edges, each requiring $\log_2 n$ bits for the origin state, $\log_2 n$ bits for the target state, and $\log_2 v$ bits for the token. Hence, the complexity is

$$C(A) \;=\; \log \binom{V}{v} \;+\; m\big(2 \log n \;+\; \log v\big). \tag{6}$$

Since $A$ changes with the problem instance, the model must synthesize in its hidden states an in-context representation of complexity $C(A)$ to predict future tokens effectively.

### 3.1.1 BORING VS. INTERESTING TASKS

We first investigate how the PHi loss differs among tasks of varying complexity. Concretely, we train our Transformer and LSTM models on a mixture of four tasks: **(1) Retrieval of Memorized Subsequences:** A small, fixed set of random subsequences is repeatedly used during training and testing. Each overall sequence is a random concatenation of these subsequences. The model can simply

memorize this small set in its static parameters and requires almost no in-context computation. **(2) Retrieval of Memorized Programs:** A small set $\mathcal{A}$ of automata (PFAs) is fixed during training and testing. Each sequence is generated by an automaton in $\mathcal{A}$. Again, the model needs only to identify which automaton from $\mathcal{A}$ is being used and then rely on its memorized representation (minimal in-context inference). **(3) In-Context Language Learning:** Each sequence is generated from a new, unfamiliar PFA $A$. The model must learn $A$ in-context, effectively synthesizing a program of complexity $C(A)$. This is a non-trivial in-context learning task. **(4) Random Examples:** Each sequence consists of random tokens with no structure to learn. While next-token prediction is difficult (leading to high loss), no meaningful in-context computation is required.

Only Task 3 demands truly interesting in-context computation: The model must discover the PFA's structure to predict future tokens effectively. By contrast, Tasks 1–2 can be solved by looking up a memorized routine, and Task 4 is just uniform noise.

Figure 3 shows the next-token prediction loss for both architectures. As expected, Tasks 1 and 2 yield low loss, while Task 4 is the highest due to random sequences. Task 3 has an intermediate loss. Note, however, that *intermediate loss alone* does not clarify whether the model has learned an intricate structure or is simply mixing easy and random components. In Figure 4 we can observe that we have a high PHi loss only for Task 3. This is in accordance with our claim that hidden states are difficult to predict when a complex program is generated in-context (Task 3), but not when easy tasks are being performed (Tasks 1, 2 & 4). These results are consistent for both the Transformer and the LSTM model.

### 3.1.2 SIMPLE VS. COMPLEX TASKS

Here, we focus exclusively on in-context language learning (Task 3 from Section 3.1.1) to investigate whether more complex tasks lead to higher PHi loss. Recall that the complexity of a stochastic language can be quantified by the code-length $C(A)$ of the corresponding PFA (see Eq. 6). Intuitively, larger or more intricate automata require more complex "in-context programs" to achieve low next-token prediction errors. The previous experiment (3.1.1) demonstrated that "boring" tasks (memorized or random) lead to more compressible hidden states (i.e., lower PHi loss), whereas "interesting" tasks (requiring non-trivial in-context learning) exhibit higher PHi loss. Now we ask: Within the realm of in-context learning itself, does the degree of complexity correlate with the magnitude of the PHi loss?

We again train our Transformer model on PFAs of varying complexity. Then we test the trained model on 1000 problem instances, each based on a different unfamiliar PFA. For each sequence, we track token-wise PHi loss as well as next-token prediction loss. Because more complex PFAs also tend to have higher irreducible uncertainty (and thus higher next-token loss), we want to show that PHi loss explains task complexity *beyond* next-token loss. To accomplish this, we bin tokens by their next-token prediction loss and compare the average PHi loss across sequences of different complexity levels. Figure 5 displays the token-wise PHi loss versus binned next-token losses, stratified by PFA complexity (grouped into 10 levels, from 1 (*simple*) to 10 (*complex*)). We see a clear trend: even after controlling for next-

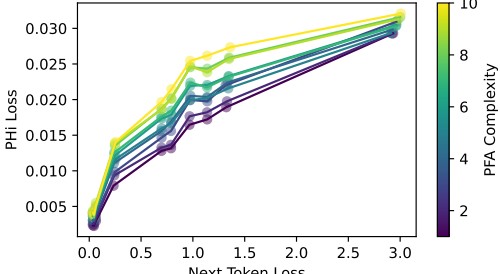

Figure 5: Token-wise PHi loss (y-axis) versus binned next-token losses (x-axis), stratified by PFA complexity (color, grouped into 10 levels, from 1 (*simple*) to 10 (*complex*)). Across all next-token prediction losses, more complex PFAs result in a higher PHi loss.

token loss, more complex PFAs yield systematically higher PHi loss. Similarly, a partial correlation analysis (controlling for mean next-token loss) reveals a significant relationship between sequence-wise mean PHi loss and PFA complexity ($r = 0.37$, with a $95\%$ confidence interval of $[0.32, 0.43]$). These findings confirm that PHi loss tracks the amount of in-context computation needed, beyond what is implied by simple token-level uncertainty.

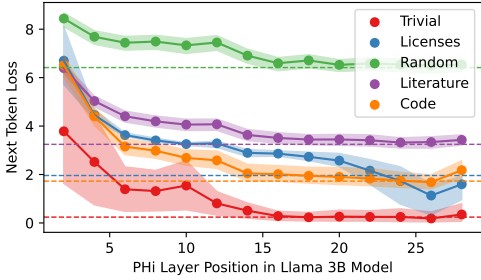 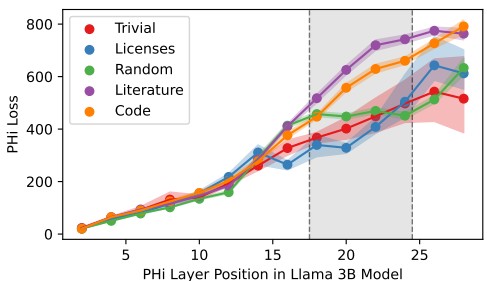

Figure 6: Next token prediction loss in a pre-trained Llama 3B model with the PHi layer inserted at different layer positions. Dotted lines show the performance of the base model without PHi layer. Note that inserting it in the lower layers causes significant degradation in the model quality because of posterior collapse, most likely due to posterior collapse. We see no correlation between next token prediction loss and the complexity of the tasks.

Figure 7: PHi loss for different tasks at different layers of a pretrained Llama 3B model. For layers 18-24 (the highlighted region), we can see the most differentiation between PHi losses for different tasks. In these layers, tasks that are intuitively "uninteresting", like predicting random sequences, memorized licenses, or trivial, memorized tasks, have a lower PHi loss compared to interesting tasks like code or literature prediction.

## 3.2 Experiments with Large Language Models

Next, we investigate whether our findings generalize to LLMs that have already been pre-trained on extensive corpora. The model weights are held fixed, and we insert and train a single PHi layer within the existing architecture. Specifically, we use the instruction-tuned "Llama 3.2" model with 3B parameters (Dubey et al., 2024), and train a single PHi layer for 10,000 steps on a mixture of reasoning and natural-language data (see Appendix C for details).

Unlike training from scratch, inserting a PHi layer into a pre-trained model raises design questions: Where in the model (which layer) do we impose the information bottleneck? The pre-trained model did not "expect" an internal bottleneck, some layers might not be able to accommodate it. Furthermore, which tasks are actually easy or difficult, simple or complex to an LLM? What kinds of texts and programs does it have memorized?

### 3.2.1 Different Language Modeling Tasks

The first experiment involving LLMs seeks to address these questions. We design a test set comprising five types of sequences, with varying expectations about memorization and in-context reasoning complexity: **(a) Trivial Tasks:** Examples such as multiplication tables, counting to 50 in Spanish, or listing weekdays in January 2025. We expect the model to have these "programs" firmly memorized and easily retrievable. **(b) Memorized Licenses:** Popular licenses on GitHub that appear frequently in pre-training data, which the model can reproduce verbatim without additional in-context reasoning. **(c) Random Words:** Shuffled tokens lacking syntactic or semantic structure. Although this results in high next-token loss, there is no interesting structure to learn in-context. **(d) Unknown Literature:** Recent Project Gutenberg chapters assumed unseen by the model. Natural language, especially literary text, often contains a rich structure that benefits from deeper in-context reasoning. **(e) Unknown Code:** A private python codebase, never encountered during pre-training. Programming languages typically require the model to capture longer-range dependencies and syntax, suggesting a non-trivial in-context program.

We group tasks (a)–(c) as "boring" (no new in-context program) and tasks (d)–(e) as "interesting" (likely to require more in-context synthesis).

We insert the PHi layer at various depths in the Llama model, then measure PHi loss and its effect on the next-token prediction loss on each task. Figure 6 shows that placing the PHi layer in early layers can lead to posterior collapse: next-token loss rises significantly (the bottleneck disrupts the flow of information), yet very low PHi loss. In contrast, placing the PHi layer in the later layers yields minimal impact on next-token accuracy and avoids collapse. In our analysis and the

remaining experiments, we focus on the instances where the PHi layer is placed after layers 18-24. There, the PHi losses are most differentiated between tasks and observe no posterior collapse. Figure 7 shows that the PHi loss is indeed higher for unknown literature and code than for trivial, memorized, or random tasks. Meanwhile, the next-token loss (6) shows no consistent pattern that discriminates "interesting" tasks from "boring" ones.

### 3.2.2 SIMPLE VS. COMPLEX TASKS IN LLMS

We next explore whether we can again distinguish simple versus complex tasks using PHi loss, this time in a pre-trained LLM. We use the same Llama-based setup (Section 3.2.1), inserting a PHi layer into the upper portion of the network while keeping the rest of the weights fixed.

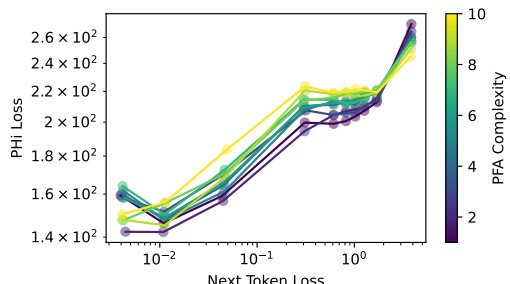

**PFA-Based Tasks.** Following a methodology similar to the Experiment 3.1.2, we instruct the LLM to generate data from a probabilistic finite automaton (PFA), then present if with an examples generated by particular problem instance. For this, we observe token-wise PHi loss and next-token prediction loss. We again bin tokens by their next-token loss and group PFA instances by complexity, $C(A)$. Figure 8

Figure 8: Token-wise PHi loss (y-axis) versus binned next-token losses (x-axis), stratified by PFA complexity (color). Results for Llama 3B with the PHi layer placed after layer 20. For most next-token loss bins, more complex PFAs result in a higher PHi loss.

shows that, much like the smaller but specialized model shown in Figure 5, complex PFAs yield higher PHi losses overall, even after controlling for next-token loss.

**Mathematical Reasoning.** We further test whether PHi loss tracks the complexity of mathematical reasoning. Specifically, we use the MATH dataset (Hendrycks et al., 2021), which consists of math problems labeled from Level 1 (easy) to Level 5 (hard), along with detailed reasoning solutions. We measure the partial correlation between PHi loss on the step-by-step solution and the difficulty of the problem and find across all tested layers a highly significant positive relationship ($r = 0.079$, with a $95\%$ confidence interval of $[0.07, 0.09]$). In line with our overall hypothesis, more difficult math problems appear to demand a more complex "in-context program" in the hidden states, leading to higher PHi loss.

### 3.2.3 CORRECT VS. ERRONEOUS RATIONALES

Finally, we examine how PHi loss relates to chains-of-thought generated by the model itself. Specifically, we look at the correctness of generated answers to questions from GSM-8k (Cobbe et al., 2021), a dataset of grade-school math problems. For each test question, we sample several solution candidates from the model, giving 8 example problems with solutions in-context and employing chain-of-thought prompting. From these candidates, we assemble random pairs of rationales for each problem, with one of the generated rationales leading to a correct final answer and the other one to an incorrect answer.

Figure 9 shows that choosing the answer for which the rationale has higher PHi loss leads to a significantly increased chance of picking the correct one (the probability of picking the correct answer by random guessing in this setting is of course 50%). It should be noted that when, from all solution pairs, we always pick the ones with *lower* mean next token prediction loss, our chance of being correct is 71%. This is surprising, since in general, PHi loss and next token prediction loss are positively correlated. It suggests that the two losses capture different important aspects of the generated rationale: A good solution to a mathematical question should be coherent (low next token loss) *and* interesting (high PHi loss). Also here, we perform a partial correlation analysis and find that PHi loss is highly predictive for correctness when controlling for next token prediction loss, across all tested PHi layer positions. Additionally, we construct a subset of "counterintuitive" questions. These are questions for which the more likely answer (the one with lower next token prediction loss) is wrong.

Even for these question, we observe the increased chance of choosing the correct answer when selecting the one with higher PHi loss. These findings suggest that, for unfamiliar or challenging math problems, the model must synthesize a more complex in-context program (and hence incur higher PHi loss) to achieve correct solutions.

We also perform the same experiments on the MATH dataset, previously used in Section 3.2.2. There, we obtain very similar results (see Appendix D.3). The classification of questions by difficulty allows for a more granular analysis. This breakdown reveals an intriguing pattern in the counterintuitive subset: For easy questions, high PHi loss does not correlate with correct rationales, whereas for difficult questions, there is a strong positive correlation between high PHi loss and correctness. We propose the following tentative explanation: In difficult counterintuitive questions, the model tends to generate misleading rationales that are overly simplistic. Arriving at the correct answer requires "effort," which is reflected in the

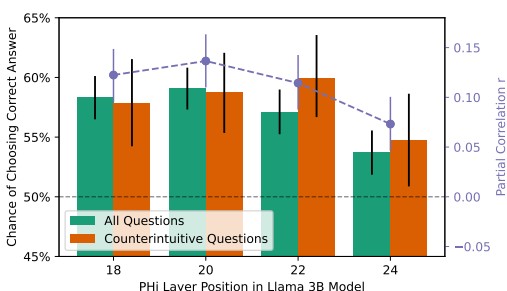

Figure 9: Chance of choosing the correct answer to mathematical reasoning questions from the GSM-8k dataset when selecting the one with the higher PHi loss between two option (one correct one false). The step-by-step rationales are generated by the model itself. "Counterintuitive" are only those questions for which the answer with the lower next token prediction loss is wrong. In purple, the partial correlation r between PHi loss and answer correctness—controlled for next token prediction loss—is shown. Answers with high PHi loss are clearly more likely to be correct.

PHi loss. In contrast, for easy counterintuitive questions, failure is not due to oversimplified rationales but might stem for example from genuine misconceptions within the model. Put differently, for difficult questions, we expect the correct answer to be "interesting"—demanding complex reasoning—whereas for easy questions, this is not necessarily the case.

## 4 RELATED WORK

Early work of hidden state prediction includes the "chunker" RNN (Schmidhuber, 1992) . Recently, self-predictive representations have gained attention in the context of reinforcement learning (Schwarzer et al., 2020; Guo et al., 2020; Ni et al., 2024). Variational auto-encoders and the reparametrization trick have introduced by Kingma & Welling (2014) and Rezende et al. (2014). Structured priors that are closer to our autoregressive prior have been proposed (Johnson et al., 2016; Li & Mandt, 2018). Information bottlenecks and autoencoders play a substantial role in interpretability research (Tishby & Zaslavsky, 2015; Jiang et al., 2020). A more detailed discussion of related work can be found in Append A.

## 5 LIMITATIONS & FUTURE WORK

Although our proposed PHi loss shows a clear correlation to the intuitive notion of "interestingness", its precise connection to formal concepts such as sophistication remains an open question. One particular issue is that of copying, or redundancy in general: When random sequences are repeated, the model must pass the information through the PHi bottleneck in order to predict the second instance of a sequence, leading to increased PHi loss. An empirical demonstration of this effect can be found in Appendix D.1. However, under most definitions, such redundancy does not constitute true "interestingness". Addressing this conceptual challenge—both theoretically and empirically—remains an important direction for future work.

At present, the hidden states need many more bits to encode than the discrete tokens, most likely due to their high dimensionality. Exploring alternative forms of information bottlenecks (e.g., quantized autoencoders (van den Oord et al., 2017)) could help bring the two losses onto more directly comparable scales. Moreover, the correlation between the next token prediction loss and the PHi loss should be explored further, possibly compensating for it when computing an aggregate score.

While we observe promising results by inserting a PHi layer into a pre-trained model, we rely on heuristics for deciding where in the network to place the bottleneck. Further empirical and theoretical studies are necessary to identify optimal bottleneck positions, and it would be valuable to train large models from scratch (or at least fine-tune them) so that they can fully accommodate the new information bottleneck. For simplicity, we aggregate the PHi loss over individual tokens by taking the mean. This works well for fixed-size sequences, but for variable-length sequences, alternative aggregation methods (e.g., summation, top-$k$ averaging, or thresholding by next-token loss) could also be considered. Finally, both "interestingness" and "task complexity" are inherently difficult to define. Consequently, evaluating our approach poses unique challenges. Developing new datasets with explicit complexity gradations—aligned to our intuitive sense of interestingness— remains an important avenue for future work.

A robust measure of interestingness could prove invaluable for agents that explore the world in an open-ended fashion (Lehman & Stanley, 2008; Hughes et al., 2024). Many ultimately important tasks provide neither external rewards nor verifiable solutions. Future work should investigate using the PHi loss of a world model as an intrinsic reward signal for exploration in reinforcement learning, or as a self-supervised objective for reasoners that must learn in the absence of external feedback.

## 6  CONCLUSION

We introduce the PHi loss, a novel information-theoretic metric for quantifying the complexity or "interestingness" of in-context computation within neural sequence models. By augmenting standard architectures (e.g., Transformers or RNNs) with our self-predictive information bottleneck (the PHi layer), we can measure when models encode non-trivial structure in their hidden states. The PHi layer can be trained both from scratch and post-hoc (with pretrained models like Llama), allowing flexibility in how our method is used. Our experiments demonstrate that PHi loss correlates with meaningful notions of task complexity, such as the description length of probabilistic finite automata, and aligns well with intuitive conceptions of complexity across various tasks. We anticipate that the PHi loss could serve as a powerful objective for applications that lack direct external feedback.

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

# A    ADDITIONAL RELATED WORK

In early work on hidden-state prediction, Schmidhuber (1992) proposed using one RNN to predict not only its next input but also the hidden state of another "chunker" RNN, which operates at a higher temporal level of abstraction. However, the chunker's hidden states had no explicit incentive to be predictable by the lower-level RNN. This approach was explored in the context of a non-generative autoencoder, encouraged to produce an internal representation that is informative about its current input while at the same time being predictable from previous inputs (Schmidhuber & Prelinger, 1993). More recently, self-predictive representations have been proposed in the context of reinforcement learning to encourage state representations that capture the important structure of the environment dynamics (Schwarzer et al., 2020; Guo et al., 2020; Ni et al., 2024).

Variational autoencoders (VAEs, Kingma & Welling (2014)) are generative models that combine variational Bayesian methods and autoencoder architectures. Their basis is the parametrization trick (Kingma & Welling, 2014; Rezende et al., 2014), which allows backpropagation through stochastic sampling operations. VAEs typically use fixed, non-informative priors, although structured priors that more closely resemble our autorregressive prior have been proposed (Johnson et al., 2016; Li & Mandt, 2018).

Various forms of information bottlenecks (Tishby & Zaslavsky, 2015) and autoencoders are widely used in mechanistic interpretability literature. Jiang et al. (2020) propose using an information bottleneck for token-level attribution of BERT models (Devlin et al., 2019). Recently, sparse autoencoders (Bricken et al., 2023) are proposed to decompose features in superposition (Elhage et al., 2022), that are common in large language models. (Henderson & Fehr, 2023) uses a VAE-based method to regularize the attention (Bahdanau et al., 2015) in transformers.

In-context learning (Brown et al., 2020; Zhao et al., 2021; Liu et al., 2022) is a well-known ability of large language models that enables learning from examples presented as inputs during inference time without explicit training. This is in contrast with the standard paradigm of train and test time separation. To uncover the mechanical underpinning of such capabilities, Akyürek et al. (2024) investigate the in-context learning of formal languages. They generate finite state automata on the fly and present transitions sampled from them to the model, and the model has to infer the structure of the automata in context. The authors call this setup "in-context language learning". We adapt their setup for our experiments in Sec. 3.1.

Prequential coding (Blier & Ollivier, 2018) is proposed to measure the complexity of neural models. It measures the amount of information needed to encode a network and its training data together, starting from an untrained model, training it online, and using the improving estimates from the model to compress the training data. The authors show that this results in good compression ratios, and more compressible models tend to generalize better. Building on these findings, Elmoznino et al. (2024) shows that the next token prediction loss naturally minimizes both the training error and the prequential code length of the implicit model learned by in-context learning, which can serve as a form of Occam's razor that encourages generalization.

Our work can be seen as explicitly quantifying and minimizing an upper bound on the complexity of this implicit model (i.e., the "program") that is generated in-context to predict the next tokens. This perspective, of jointly minimizing a two-part description—the complexity of a model plus the complexity of the data given that model (Grünwald, 2007)—relates closely to a concept from algorithmic information theory called *sophistication* (Koppel, 1987; Kolmogorov, 1974). This uncomputable quantity is the complexity of the simplest model which minimizes the two part description length of some data; it captures the amount of structured, meaningful or "interesting" information contained the data (Vitányi, 2006). Sophistication is closely related to logical depth (Bennett, 1988; Antunes et al., 2017), another concept that seeks to quantify interestingness. Other views of interestingness relate it to discoverable compression progress (Schmidhuber, 1999; 2009), or indeed–as we do in this work–to information gain (Storck et al., 1995; Itti & Baldi, 2005; Herrmann et al., 2023).

## B    Details on Experiments with Fully Trained Sequence Models

We train both the Transformer and the LSTM model for $30,000$ steps using the Adam optimizer, a batch size of 16 and gradient norm clipping of 1.0. The learning rate is 0.0003, with a 500 step linear warm-up from zero and no decay. The standard next token prediction loss and PHi loss are both weighted with 1, but we the PHi loss we take the mean of the element-wise KL-Divergence for $z$, not the sum. To prevent posterior collapse, we employ a contrastive self-critic loss (Menon et al., 2022) with weighting factor of 0.1.

### Transformer Hyperparameters

The model is based on the Llama 3.2 architecture (Dubey et al., 2024)

- Number of layers: 12
- Model dimensionality: 768
- Number of attention heads: 6
- MLP intermediate size: 2048
- Position of PHi layer: After layer 6

PHi Layer:

- $z$ dimensionality: 768
- $q_\psi$: Linear transform for $\mu$ and $\sigma$ each
- $a_\xi$: Linear transform
- $p_\chi$: One transformer layer like the ones in the rest of the model

### LSTM Hyperparameters

- Number of layers: 2
- Model dimensionality: 768
- Position of PHi layer: After layer 1

PHi Layer:

- $z$ dimensionality: 768
- $q_\psi$: Linear transform for $\mu$ and $\sigma$ each
- $a_\xi$: Linear transform
- $p_\chi$: One Llama transformer layer with a dimensionality of 768, 6 attention heads, and an intermediate MLP size of 2048

### Training and Testing Data Generation

We follow Akyürek et al. (2024) for the exact setup of the in-context langauge learning tasks. The parameters of the PFAs are randomly sampled from the following ranges:

- Number of states: $3 - 12$
- Number of edges per state: $1 - 4$
- Vocabulary size: $4 - 18$
- Example length: $10 - 50$

Each sequence consists of $10 - 20$ examples from one PFA. To robustify training, for half of the examples, we randomly perturb $20\%$ of the tokens. During testing we use no perturbation.

As data for the four different tasks in experiment 3.1.1 we use for

- Task 1: 10 randomly generated examples
- Task 2: Examples procedurally generated from one of 10 fixed PFAs
- Task 3: Examples procedurally generated from a newly sampled PFA
- Task 4: Randomly sampled examples

Each sequence is uniformly sampled from one of the tasks.

For experiment 3.1.2, we train a transformer with the same hyperparameters and training conditions as described above. However, here we train exclusively on data from Task 3. This is to avoid any biases due to the memorized PFAs from task 2.

## C  DETAILS ON EXPERIMENT WITH PRE-TRAINED LLMS

The models involved in all of the experiments have the same hyperparameters and training data. We fix the pre-trained instruction-tuned Llama 3B model's parameters and only optimize the weights of the PHi layer. They are trained for $10,000$ steps using the Adam optimizer, a batch size of 2 and gradient norm clipping of 1.0. The learning rate is 0.0001, with no warm-up or decay. No contrastive self-critic loss is used. One model is trained with the PHi layer positioned after layer 2, 4, 6, 8, 10, 12, 14, 16, 18, 20, 22, 24, 26, or 28, respectively.

PHi Layer:

- $z$ dimensionality: 3072
- $q_\psi$: Linear transform for $\mu$ and $\sigma$ each
- $a_\xi$: Linear transform
- $p_\chi$: One transformer layer like the ones in the rest of the model

The models are trained on a mixture of natural language data from the SlimPajama dataset, the MATH training set and the GSM-8k training set.

### EVALUATION DATA

In section 3.2.1, we use the following data:

- Trivial tasks:
    - *Write the multiplication table from 1 to 10.*
    - *List all the dates in January 2025 with their day of the week.*
    - *Convert the numbers 1 to 50 into binary.*
    - *Write the numbers 1 to 50 in Spanish.*
    - *Repeat the string 'RJGHDTSL' 64 times.*

- Licenses tasks:
    - *Generate the MIT license.*
    - *Generate the Apache 2.0 license.*
    - *Generate the GNU General Public License 2.0.*

- Random tasks: Examples from the SlimPajama dataset where the word order is randomly shuffled.

- Literature tasks:
    - *The beginning of 'Nigel Browning' by Agnes Giberne.*
    - *The beginning of 'Binney the Beaver' by Lucy Ellen Guernsey.*
    - *The beginning of 'The Red-Hot Dollar' by H.D. Umbstaetter*
    - *The beginning of 'Hooking a Sky Ride' by Dan Morrissey.*
    - *The beginning of 'Flying Down a Rainbow' by Homer King Gordon.*
    - *The beginning of 'Boots - A story of the Sierra Paru' by Murray Leinster.*
    - *The beginning of 'The Radio Cop' by Vic Whitman.*
    - *The beginning of 'The Trap' by Murray Leinster.*
    - *The beginning of 'The Cradle of the Deep' by Joan Lowell.*
    - *The beginning of 'Fix Bayonets!' by John W. Thomason, Jr.*

- Code tasks: Files from a private python code base that deals mainly with symbolic processing of music data.

# D    ADDITIONAL RESULTS

## D.1    PHI LOSS AND COPYING

To highlight the conceptual issue of interestingness and redundancy, we extend the experiments from Section 3.1.1 by introducing an additional task: copying. Similar to the random task, each sequence consists of previously unseen random subsequences. However, in this case, every subsequence appears twice within the same context. On its second occurrence, the model should be able to predict it almost perfectly via context retrieval. To do that, the information has to cross the PHi bottleneck, resulting in high PHi loss. Figures 10 and 11 show the next token loss and the PHi loss for fully trained models across all five tasks, including copying. For the transformer, the PHi loss in the copy task remains significantly lower than in in-context language learning. Still, it is debatable whether the copying task should be regarded as at all interesting. Note that this version of the copying task, with an exact 1-to-1 ratio of new random sequences to copies, maximizes the PHi loss. Increasing the proportion of either copies or random data would reduce the PHi loss per token, as the total information passing through the bottleneck is constrained by the lesser of the two components.

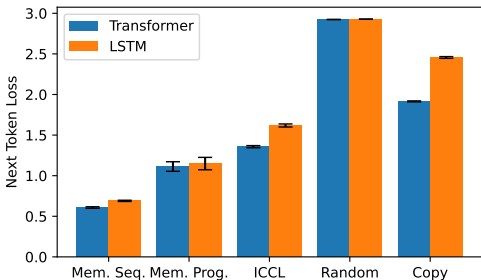
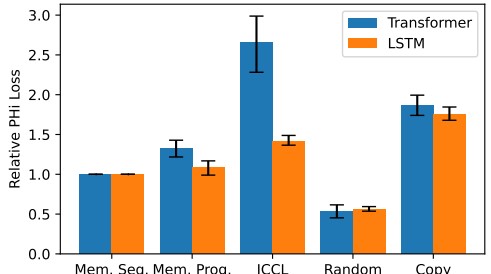

Figure 10: Token-wise PHi loss versus binned next-token losses, stratified by PFA complexity (color). Normalized across each bin. Results from a fully trained Transformer model. Across all next-token prediction losses, more complex PFAs result in a higher PHi loss.

Figure 11: Token-wise PHi loss versus binned next-token losses, stratified by PFA complexity (color). Normalized across each bin. Results from Llama 3B model with the PFA layer placed after layer 20. Also here, across most next-token prediction losses, more complex PFAs result in a higher PHi loss.

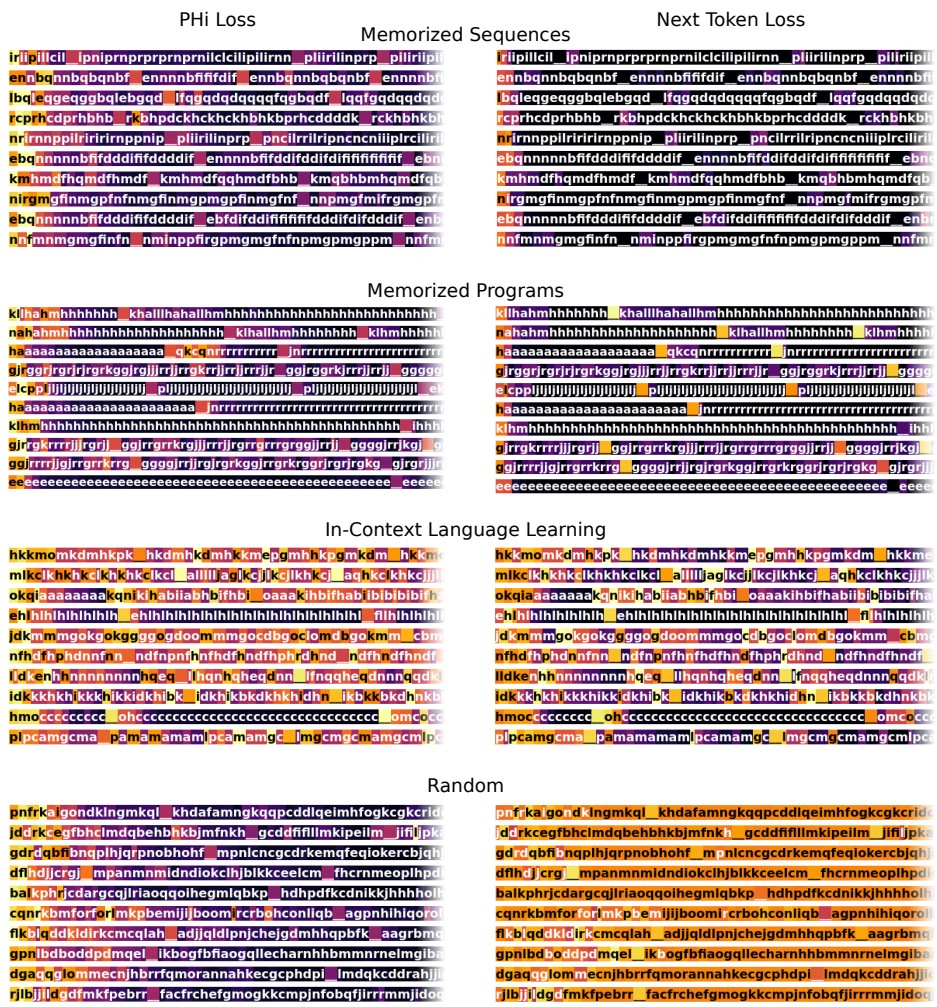

Figure 12: Examples of the four different tasks discussed in Section 3.1.1, colored by the PHi and next token loss from a Transformer model trained from scratch. We see clearly that the in-context language learning task leads to higher PHi values, especially in the later parts of the sequences.

## D.2 PFA TASK EXAMPLES AND NORMALIZED COMPLEXITY

Figure 12 shows concrete examples of in-context language learning tasks, along with memorized sequence, memorized programs and random data. Figures 13 and 14 show alternative versions of Figures 5 and Figure 8, respectively. Here, the PHi losses are normalized across each next token prediction loss bin.

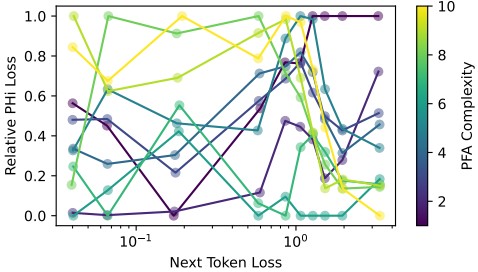
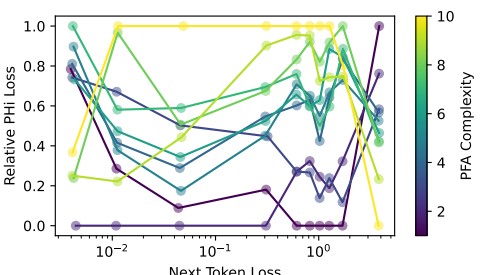

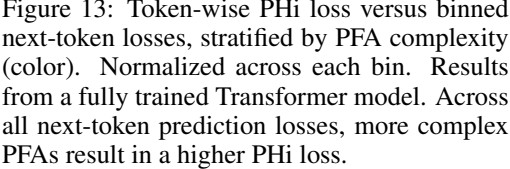

Figure 13: Token-wise PHi loss versus binned next-token losses, stratified by PFA complexity (color). Normalized across each bin. Results from a fully trained Transformer model. Across all next-token prediction losses, more complex PFAs result in a higher PHi loss.

Figure 14: Token-wise PHi loss versus binned next-token losses, stratified by PFA complexity (color). Normalized across each bin. Results from Llama 3B model with the PFA layer placed after layer 20. Also here, across most next-token prediction losses, more complex PFAs result in a higher PHi loss.

### D.3 CORRECT VS. ERRONEOUS RATIONALES FOR THE MATH DATASET

These results extend those presented in Section 3.2.3. Figure 15 shows the chance of choosing the correct answer for MATH dataset problems when picking the one with higher PHi loss, both across all answer pairs and for counterintuitive ones specifically. We use a chain-of-thought prompt but, following common practice for this dataset, provide no in-context examples. Once again, we find that responses with high PHi loss are significantly more likely to be correct. Compared to results on GSM-8k, selecting the response with the lowest next token loss is less predictive of correctness (only ca. $52\%$). Figure 16 further breaks down the MATH dataset by difficulty. Notably, for counterintuitive questions, a strong correlation emerges between high PHi loss and correctness in difficult problems, whereas this relationship is absent in easier ones. While this finding warrants deeper analysis, we propose the following explanation: Recall that we define counterintuitive questions as those where the response with lower next-token loss is incorrect. Such questions may be counterintuitive for various reasons. One possibility is that they contain a tempting but ultimately incorrect simplistic solution. In this case, high PHi loss—signaling greater cognitive effort—may indicate that the model is resisting this temptation. For difficult problems, this factor may dominate. Conversely, for easier problems, counterintuitiveness may stem from factual errors, which are not necessarily linked to PHi loss.

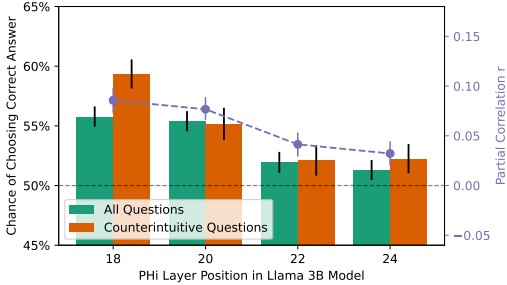

Figure 15: Similar to Figure 9, but for the MATH dataset. Chance of choosing the correct answer when selecting the one with the higher PHi loss between a correct and a wrong option. "Counterintuitive" are only those questions for which the answer with the lower next token prediction loss is wrong. In purple, the partial correlation r between PHi loss and answer correctness—controlled for next token prediction loss—is shown. Answers with high PHi loss are clearly more likely to be correct.

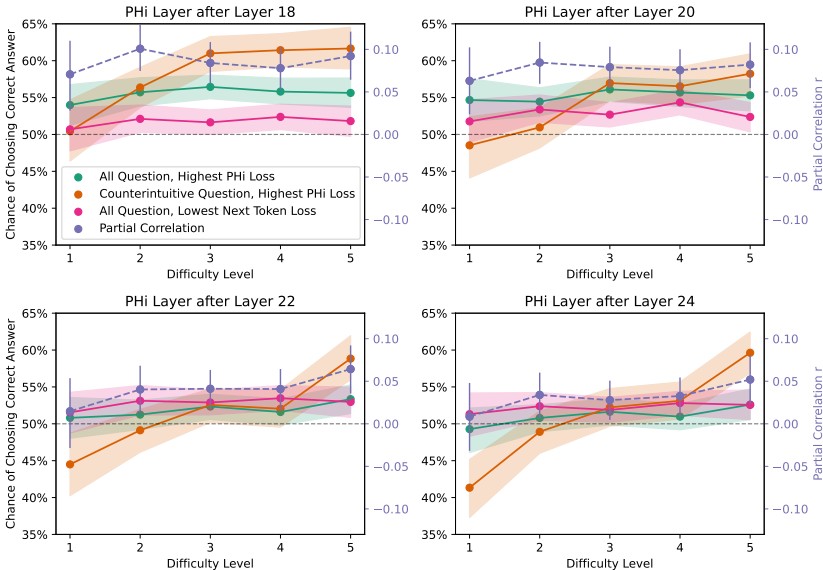

Figure 16: Detailed view of rationales generated for the MATH dataset, separated by the difficulty of the questions. Curves show (green) the chance of choosing the correct answer when selecting the one with highest PHi loss among all answer pairs, (orange) among the counterintuitive pairs, (pink) among all pairs when selecting the answer with lowest next token loss, and (purple) the partial correlation between PHi loss and answer correctness, controlled for next token loss. We see a strong correlation between the correctness of the rationale and PHi loss, especially for difficult counterintuitive questions. For easy counterintuitive question, this relationship does not exist.

## D.4 HISTOGRAMS FOR NEXT TOKEN AND PHI LOSS

Figure 17 provides a more detailed view of the results from Section 3.1.1 in the form of two-dimensional histograms, with next-token loss and PHi loss as the axes. Figure 18 presents similar results for models trained also with the copying task. As shown in the top-right histogram, the Transformer model exhibits a distinct cluster of copied sequences characterized by low next-token loss but high PHi loss. In contrast, the LSTM model does not display this cluster, likely because it struggles to adequately perform the copying task. Copying is known to be challenging for LSTMs—see, for example, Graves et al. (2014).

Figure 19 presents analogous histograms for the Llama 3B model on tasks from Section 3.2.1.

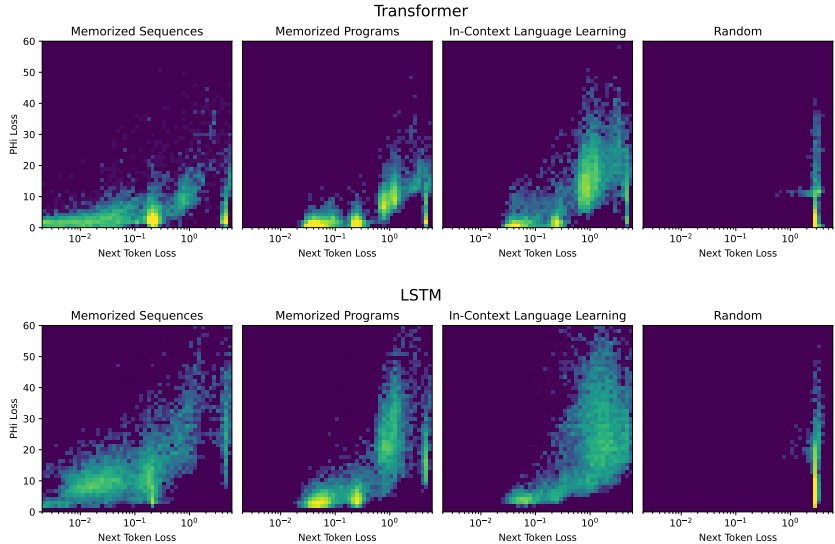

Figure 17: Two-dimensional histogram of tokens from the four different tasks presented in Section 3.1.1, PHi loss (y-axis) versus next token loss (x-axis). For memorized sequences, tokens are concentrated on the lower left, whereas for random data, they are clustered in the lower right corner. Only in-context language learning yields a significant amount of tokens that are high up on the y-axis. 10,000 tokens per task from a Transformer, and from an LSTM model.

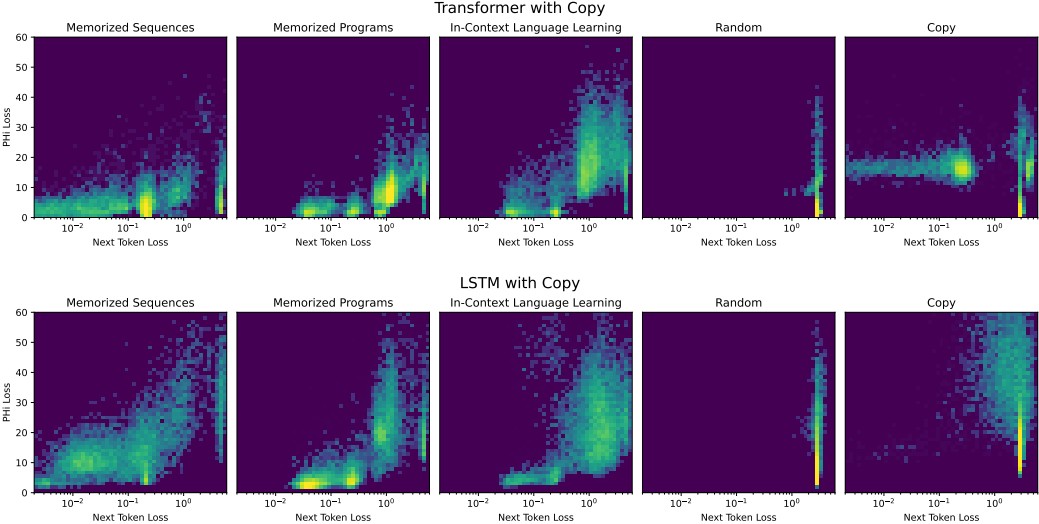

Figure 18: Similar to Figure 17, but including the copying task. For the transformer model, we can see that copied sequences have low next token loss, but relatively high PHi loss.

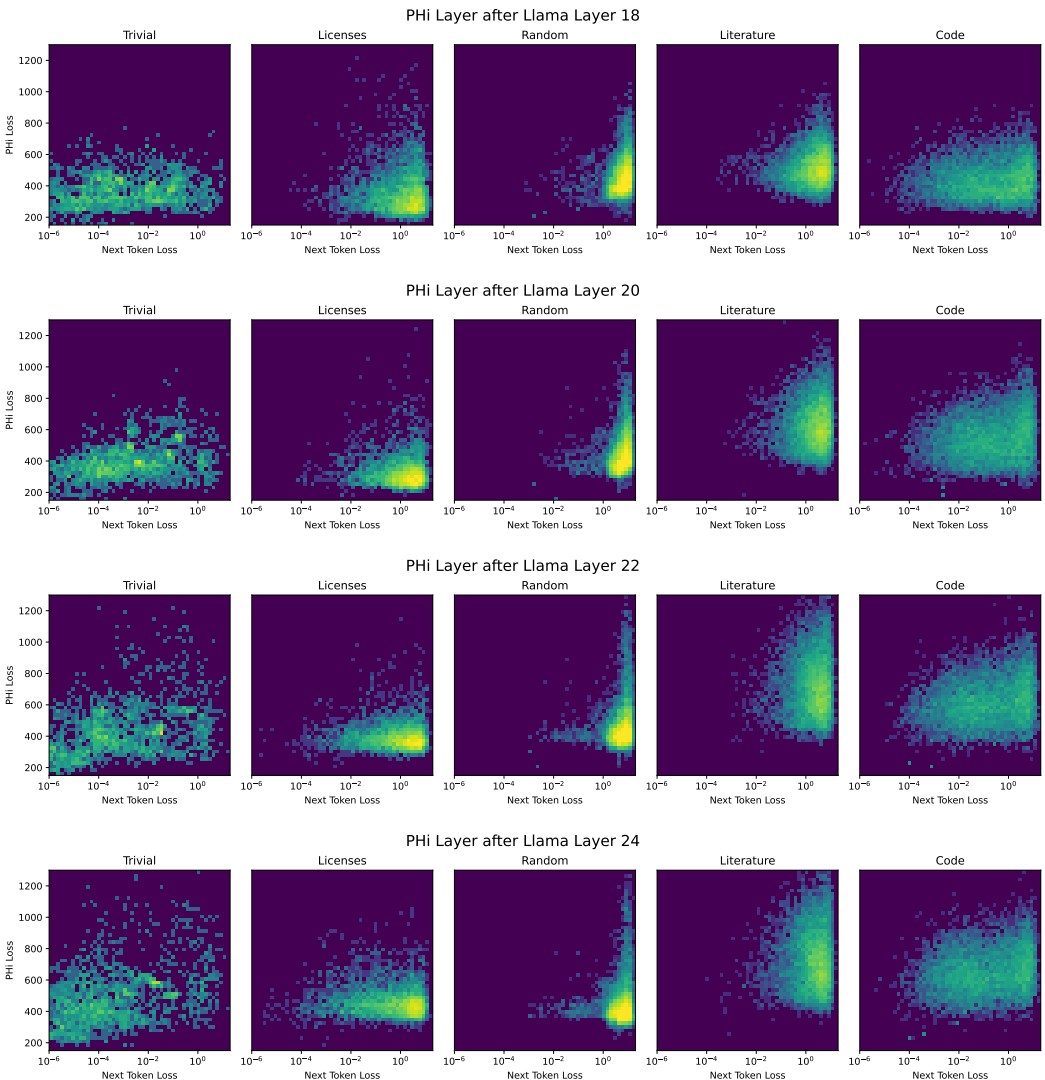

Figure 19: Two-dimensional histograms of tokens from the five types of data described in Section 3.2.1. For Literature and Code we see more token spreading to the upper, high PHi loss part of the histogram. Data shown for the PHi layer located after layers 18, 20, 22 and 24 of the Llama 3B model.

