# OpenReview forum: "Measuring In-Context Computation Complexity via Hidden State Prediction"
_ICLR.cc/2025/Workshop/BuildingTrust — BuildingTrust_

### Official Review · Reviewer_Qwkj · 2025-02-21
**PHi Loss**

**Rating:** 8
**Confidence:** 4

**Review:**

The authors propose PHi Loss, which measures the model's ability to predict its own next hidden state, as a metric to identify when a sequence model is doing interesting in-context computations. The authors found that PHi Loss outperform next token prediction loss for a wide range of datasets and models.

- Questions & Comments
1) Perhaps it would be better to define Phi Loss as the inverse of KL divergence, since we usually associate lower loss to be better. In Figure 3, random sequence currently has the lowest loss.
2) In Figure 6, it would be interesting to plot the difference between consecutive PHi loss, which would essentially loss how much interesting computation is performed in each layer. It seems like this derivative plot will look like a parabola with the peak at the middle layer. This is consistent with literature showing that most interesting computations (e.g. understanding high-level semantic concepts) are performed in the middle layer.

---

### Official Review · Reviewer_kENj · 2025-02-24

**Rating:** 9
**Confidence:** 4

**Review:**

### Summary
This paper proposes a novel metric, the Prediction of Hidden States (PHi) loss, to measure the complexity of in-context computation in large language models (LLMs). Traditional next-token prediction loss fails to distinguish between meaningful computation and trivial memorization, prompting the authors to introduce PHi loss, which measures how well a model can predict its own future hidden states.
To implement this, the authors introduce a PHi layer, which acts as an information bottleneck in the model's residual stream. The key idea is that the KL divergence between the posterior and the prior in the PHi layer quantifies the amount of new information learned at each step. The authors validate their approach with various experiments, showing that PHi loss correlates with task complexity in in-context language learning, mathematical reasoning, and self-generated reasoning chains.

### Strongness
- Unlike traditional next-token loss, PHi loss provides insight into how much non-trivial computation is occurring within a model’s hidden states.It is architecture-agnostic and can be integrated into different neural architectures, including Transformers and RNNs.
- The authors demonstrate the effectiveness of PHi loss across in-context language learning, mathematical problem-solving, and reasoning tasks.Their findings indicate that higher PHi loss correlates with higher task complexity, making it a potentially useful metric for understanding model reasoning.
- The study extends beyond training from scratch by incorporating PHi layers into pretrained models like LLaMA-3B, providing a feasible post-hoc method for analyzing existing models.
- PHi loss could be used to improve interpretability, guide model training, or even serve as an intrinsic reward in reinforcement learning setups.

### Questions
- A key question is whether optimizing for lower PHi loss actually leads to better model performance.

---

### Official Review · Reviewer_sfyP · 2025-03-02
**Proposing a very interesting and effective way to quantify "interesting" computation in the in-context inference**

**Rating:** 8
**Confidence:** 2

**Review:**

The paper proposes a way to measure how much "interesting" computation is done during in-context sequence prediction. Instead of quantifying this using next-token prediction loss, which would have high loss even when the sequence is random (indicating no in-context learning is required), they propose predicting the next hidden state.

They insert a layer within the transformer to use the latent state as an information bottleneck and introduce a module to predict the next latent state based on the history of latents. Then, they compute the KL divergence between the prior (based on latent state history) and the posterior (conditioned on the next token's hidden state) and call it PHi loss.

Experimental results compare "boring" tasks, such as tasks focused on memorization and random sequences, with "interesting" tasks, such as literature and coding. Their quantification method shows that PHi loss is indeed higher for interesting tasks and lower for boring tasks. They also conduct a control experiment using probabilistic finite automaton sequences, demonstrating that more complex structures correspond to higher PHi loss.

This is a very interesting and effective way to quantify in-context learning computation, and they further show that it can be applied to a 3B parameter pre-trained model as well.

---

### Decision · Program_Chairs · 2025-03-04

Accept